# Diverse Trajectory Forecasting with Determinantal Point Processes

**Ye Yuan, Kris M. Kitani**
Robotics Institute
Carnegie Mellon University
{yyuan2,kkitani}@cs.cmu.edu

## Abstract

The ability to forecast a set of likely yet *diverse* possible future behaviors of an agent (*e.g.*, future trajectories of a pedestrian) is essential for safety-critical perception systems (*e.g.*, autonomous vehicles). In particular, a set of possible future behaviors generated by the system must be diverse to account for all possible outcomes in order to take necessary safety precautions. It is not sufficient to maintain a set of the most likely future outcomes because the set may only contain perturbations of a dominating single outcome (major mode). While generative models such as variational autoencoders (VAEs) have been shown to be a powerful tool for learning a distribution over future trajectories, randomly drawn samples from the learned implicit likelihood model may not be diverse – the likelihood model is derived from the training data distribution and the samples will concentrate around the major mode of the data. In this work, we propose to learn a diversity sampling function (DSF) that generates a diverse yet likely set of future trajectories. The DSF maps forecasting context features to a set of latent codes which can be decoded by a generative model (*e.g.*, VAE) into a set of diverse trajectory samples. Concretely, the process of identifying the diverse set of samples is posed as DSF parameter estimation. To learn the parameters of the DSF, the diversity of the trajectory samples is evaluated by a diversity loss based on a determinantal point process (DPP). Gradient descent is performed over the DSF parameters, which in turn moves the latent codes of the sample set to find an optimal set of diverse yet likely trajectories. Our method is a novel application of DPPs to optimize a set of items (forecasted trajectories) in continuous space. We demonstrate the diversity of the trajectories produced by our approach on both low-dimensional 2D trajectory data and high-dimensional human motion data. (Video[1])

## 1 Introduction

Forecasting future trajectories of vehicles and human has many useful applications in autonomous driving, virtual reality and assistive living. What makes trajectory forecasting challenging is that the future is uncertain and multi-modal – vehicles can choose different routes and people can perform different future actions. In many safety-critical applications, it is important to consider a diverse set of possible future trajectories, even those that are less likely, so that necessary preemptive actions can be taken. For example, an autonomous vehicle should understand that a neighboring car can merge into its lane even though the car is most likely to keep driving straight. To address this requirement, we need to take a generative approach to trajectory forecasting that can fully characterize the multi-modal distribution of future trajectories. To capture all modes of a data distribution, variational autoencoders (VAEs) are well-suited generative models. However, random samples from a learned VAE model with Gaussian latent codes are not guaranteed to be diverse for two reasons. First, the sampling procedure is stochastic and the VAE samples can fail to cover some minor modes even with a large number of samples. Second, since VAE sampling is based on the implicit likelihood function encoded in the training data, if most of the training data is centered around a specific mode while other modes have less data (Fig. 1 (a)), the VAE samples will reflect this bias and concentrate around the major mode (Fig. 1 (b)). To tackle this problem, we propose to learn a diversity sampling function (DSF) that can reliably generate a diverse set of trajectory samples (Fig. 1 (c)).

---

[1] https://youtu.be/5i71SU_IdS4

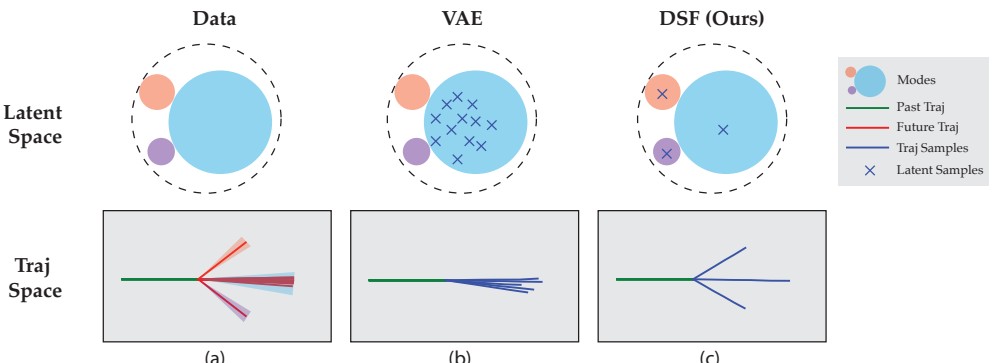

Figure 1: A toy trajectory forecasting example. (a) The three modes (pink, blue, purple) of the future trajectory distribution are shown in both the trajectory space and the latent space of a learned VAE model. The data distribution is imbalanced, where the blue mode has most data and covers most of the latent space. (b) Random samples from the VAE only cover the major (blue) mode. (c) Our proposed DSF generates a diverse set of future trajectories covering all three modes.

The proposed DSF is a deterministic parameterized function that maps forecasting context features (*e.g.,* past trajectories) to a set of latent codes. The latent codes are decoded by the VAE docoder into a set of future trajectory samples, denoted as the DSF samples. In order to optimize the DSF, we formulate a diversity loss based on a determinantal point process (DPP) (Macchi, 1975) to evaluate the diversity of the DSF samples. The DPP defines the probability of choosing a random subset from the set of trajectory samples. It models the negative correlations between samples: the inclusion of a sample reduces the probability of including a similar sample. This makes the DPP an ideal tool for modeling the diversity within a set. In particular, we use the expected cardinality of the DPP as the diversity measure, which is defined as the expected size of a random subset drawn from the set of trajectory samples according to the DPP. Intuitively, since the DPP inhibits selection of similar samples, if the set of trajectory samples is more diverse, the random subset is more likely to select more samples from the set. The expected cardinality of the DPP is easy to compute and differentiable, which allows us to use it as the objective to optimize the DSF to enable diverse trajectory sampling.

Our contributions are as follows: (1) We propose a new forecasting approach that learns a diversity sampling function to produce a diverse set of future trajectories; (2) We propose a novel application of DPPs to optimize a set of items (trajectories) in continuous space with a DPP-based diversity measure; (3) Experiments on synthetic data and human motion validate that our method can reliably generate a more diverse set of future trajectories compared to state-of-the-art generative models.

## 2 RELATED WORK

**Trajectory Forecasting** has recently received significant attention from the vision community. A large portion of previous work focuses on forecasting 2D future trajectories for pedestrians (Kitani et al., 2012; Ma et al., 2017; Ballan et al., 2016; Xie et al., 2013) or vehicles (Jain et al., 2016a). Some approaches use deterministic trajectory modeling and only forecast one future trajectory (Alahi et al., 2016; Yagi et al., 2018; Robicquet et al., 2016). As there are often multiple plausible future trajectories, several approaches have tried to forecast distributions over trajectories (Lee et al., 2017; Galceran et al., 2015; Gupta et al., 2018). Recently, Rhinehart et al. (2018; 2019) propose a generative model that can accurately forecast multi-modal trajectories for vehicles. Soo Park et al. (2016) also use egocentric videos to predict the future trajectories of the camera wearer. Some work has investigated forecasting higher dimensional trajectories such as the 3D full-body pose sequence of human motions. Most existing work takes a deterministic approach and forecasts only one possible future motion from past 3D poses (Fragkiadaki et al., 2015; Butepage et al., 2017; Li et al., 2017; Jain et al., 2016b), static images (Chao et al., 2017; Kanazawa et al., 2018) or egocentric videos (Yuan and Kitani, 2019). Differently, some probabilistic approaches (Habibie et al., 2017; Yan et al., 2018) use conditional variational autoencoders (cVAEs) to generate multiple future motions. In constrast to previous work, our approach can generate a *diverse* set of future motions with a limited number of samples.

**Diverse Solutions** have been sought after in a number of problems in computer vision and machine learning. A branch of these methods aiming for diversity stems from the M-Best MAP problem (Nilsson, 1998; Seroussi and Golmard, 1994), including diverse M-Best solutions (Batra et al., 2012) and multiple choice learning (Guzman-Rivera et al., 2012; Lee et al., 2016). Alternatively, previous work has used submodular function maximization to select a diverse subset of garments from fashion images (Hsiao and Grauman, 2018). Determinantal point processes (DPPs) (Macchi, 1975; Kulesza et al., 2012) are efficient probabilistic models that can measure both the diversity and quality of items in a subset, which makes it a natural choice for the diverse subset selection problem. DPPs have been applied for document and video summarization (Kulesza and Taskar, 2011; Gong et al., 2014), recommendation systems (Gillenwater et al., 2014), object detection (Azadi et al., 2017), and grasp clustering (Huang et al., 2015). Elfeki et al. (2018) have also used DPPs to mitigate mode collapse in generative adversarial networks (GANs). The work most related ours is (Gillenwater et al., 2014), which also uses the cardinality of DPPs as a proxy for user engagement. However, there are two important differences between our approach and theirs. First, the context is different as they use the cardinality for a subset selection problem while we apply the cardinality as an objective of a continuous optimization problem in the setting of generative models. Second, their main motivation behind using the cardinality is that it aligns better with the user engagement semantics, while our motivation is that using cardinality as a diversity loss for deep neural networks is more stable due to its tolerance of similar trajectories, which are often produced by deep neural networks during stochastic gradient descent.

## 3 BACKGROUND

### 3.1 VARIATIONAL AUTOENCODERS

The aim of multi-modal trajectory forecasting is to learn a generative model over future trajectories. Variational autoencoders (VAEs) are a popular choice of generative models for trajectory forecasting (Lee et al., 2017; Walker et al., 2016) because it can effectively capture all possible future trajectories by explicitly mapping each data point to a latent code. VAEs model the joint distribution $p_\theta(\mathbf{x}, \mathbf{z}) = p(\mathbf{z})p_\theta(\mathbf{x}|\mathbf{z})$ of each data sample $\mathbf{x}$ (*e.g.*, a future trajectory) and its corresponding latent code $\mathbf{z}$, where $p(\mathbf{z})$ denotes some prior distribution (*e.g.*, Gaussians) and $p_\theta(\mathbf{x}|\mathbf{z})$ denotes the conditional likelihood model. To calculate the marginal likelihood $p_\theta(\mathbf{x}) = p_\theta(\mathbf{x}, \mathbf{z})/p_\theta(\mathbf{z}|\mathbf{x})$, one needs to compute the posterior distribution $p_\theta(\mathbf{z}|\mathbf{x})$ which is typically intractable. To tackle this, VAEs use variational inference (Jordan et al., 1999) which introduces an approximate posterior $q_\phi(\mathbf{z}|\mathbf{x})$ and decomposes the marginal log-likelihood as

$$\log p_\theta(\mathbf{x}) = \mathrm{KL}\left(q_\phi(\mathbf{z}|\mathbf{x})\|p_\theta(\mathbf{z}|\mathbf{x})\right) + \mathcal{L}(\mathbf{x}; \theta, \phi), \tag{1}$$

where $\mathcal{L}(\mathbf{x}; \theta, \phi)$ is the evidence lower bound (ELBO) defined as

$$\mathcal{L}(\mathbf{x}; \theta, \phi) = \mathbb{E}_{q_\phi(\mathbf{z}|\mathbf{x})}\left[\log p_\theta(\mathbf{x}|\mathbf{z})\right] - \mathrm{KL}\left(q_\phi(\mathbf{z}|\mathbf{x})\|p(\mathbf{z})\right). \tag{2}$$

During training, VAEs jointly optimize the recognition model (encoder) $q_\phi(\mathbf{z}|\mathbf{x})$ and the likelihood model (decoder) $p_\theta(\mathbf{x}|\mathbf{z})$ to maximize the ELBO. In the context of multi-modal trajectory forecasting, one can generate future trajectories from $p(\mathbf{x})$ by drawing a latent code $\mathbf{z}$ from the prior $p(\mathbf{z})$ and decoding $\mathbf{z}$ with the decoder $p_\theta(\mathbf{x}|\mathbf{z})$ to produce a corresponding future trajectory $\mathbf{x}$.

### 3.2 DETERMINANTAL POINT PROCESSES

Our core technical innovation is a method to learn a *diversity* sampling function (DSF) that can generate a diverse set of future trajectories. To achieve this, we must equip ourselves with a tool to evaluate the diversity of a set of trajectories. To this end, we make use of determinantal point processes (DPPs) to model the diversity within a set. DPPs promote diversity within a set because the inclusion of one item makes the inclusion of a similar item less likely if the set is sampled according to a DPP.

Formally, given a set of items (*e.g.,* data points) $\mathcal{Y} = \{\mathbf{x}_1, \ldots, \mathbf{x}_N\}$, a point process $\mathcal{P}$ on the ground set $\mathcal{Y}$ is a probability measure on $2^{\mathcal{Y}}$, *i.e.*, the set of all subsets of $\mathcal{Y}$. $\mathcal{P}$ is called a determinantal point process if a random subset $\boldsymbol{Y}$ drawn according to $\mathcal{P}$ has

$$\mathcal{P}_{\mathbf{L}}(\boldsymbol{Y} = Y) = \frac{\det\left(\mathbf{L}_Y\right)}{\sum_{Y \subseteq \mathcal{Y}} \det\left(\mathbf{L}_Y\right)} = \frac{\det\left(\mathbf{L}_Y\right)}{\det(\mathbf{L} + \mathbf{I})}, \tag{3}$$

where $Y \subseteq \mathcal{Y}$, $\mathbf{I}$ is the identity matrix, $\mathbf{L} \in \mathbb{R}^{N \times N}$ is the DPP kernel, a symmetric positive semidefinite matrix, and $\mathbf{L}_Y \in \mathbb{R}^{|Y| \times |Y|}$ is a submatrix of $\mathbf{L}$ indexed by elements of $Y$.

The DPP kernel $\mathbf{L}$ is typically constructed by a *similarity* matrix $\mathbf{S}$, where $\mathbf{S}_{ij}$ defines the similarity between two items $\mathbf{x}_i$ and $\mathbf{x}_j$. If we use the inner product as the similarity measure, $\mathbf{L}$ can be written in the form of a Gram matrix $\mathbf{L} = \mathbf{S} = \mathbf{X}^T \mathbf{X}$ where $\mathbf{X}$ is the stacked feature matrix of $\mathcal{Y}$. As a property of the Gram matrix, $\det(\mathbf{L}_Y)$ equals the squared volume spanned by vectors $\mathbf{x}_i \in Y$. With this geometric interpretation in mind, one can observe that diverse sets are more probable because their features are more orthogonal, thus spanning a larger volume.

In addition to set diversity encoded in the similarity matrix $\mathbf{S}$, it is also convenient to introduce a *quality* vector $\mathbf{r} = [r_1, \ldots, r_N]$ to weigh each item according to some unary metric. For example, the quality weight might be derived from the likelihood of an item. To capture both diversity and quality of a subset, the DPP kernel $\mathbf{L}$ is often decomposed in the more general form:

$$\mathbf{L} = \mathrm{Diag}(\mathbf{r}) \cdot \mathbf{S} \cdot \mathrm{Diag}(\mathbf{r}). \tag{4}$$

With this decomposition, we can see that both the quality vector $\mathbf{r}$ and similarity matrix $\mathbf{S}$ contribute to the DPP probability of a subset $Y$:

$$\mathcal{P}_L(\mathbf{Y} = Y) \propto \det(\mathbf{L}_Y) = \left( \prod_{\mathbf{x}_i \in Y} r_i^2 \right) \det(\mathbf{S}_Y). \tag{5}$$

Due to its ability to capture the global diversity and quality of a set of items, we choose DPPs as the probabilistic approach to evaluate and optimize the diversity of the future trajectories drawn by our proposed diversity sampling function.

## 4 APPROACH

Safety-critical applications often require that the system can maintain a diverse set of outcomes covering all modes of a predictive distribution and not just the most likely one. To address this requirement, we propose to learn a diversity sampling function (DSF) to draw deterministic trajectory samples by generating a set of latent codes in the latent space of a conditional variational autoencoder (cVAE) and decoding them into trajectories using the cVAE decoder. The DSF trajectory samples are evaluated with a DPP-based diversity loss, which in turn optimizes the parameters of the DSF for more diverse trajectory samples.

Formally, the future trajectory $\mathbf{x} \in \mathbb{R}^{T \times D}$ is a random variable denoting a $D$ dimensional feature over a future time horizon $T$ (*e.g.,* a vehicle trajectory or a sequence of humanoid poses). The context $\boldsymbol{\psi} = \{\mathbf{h}, \mathbf{f}\}$ provides the information to infer the future trajectory $\mathbf{x}$, and it contains the past trajectory $\mathbf{h} \in \mathbb{R}^{H \times D}$ of last $H$ time steps and optionally other side information $\mathbf{f}$, such as an obstacle map. In the following, we first describe how we learn the future trajectory model $p_\theta(\mathbf{x}|\boldsymbol{\psi})$ with a cVAE. Then, we introduce the DSF and the DPP-based diversity loss used to optimize the DSF.

### 4.1 LEARNING A CVAE FOR FUTURE TRAJECTORIES

In order to generate a diverse set of future trajectory samples, we need to learn a generative trajectory forecasting model $p_\theta(\mathbf{x}|\boldsymbol{\psi})$ that can cover all modes of the data distribution. Here we use cVAEs (other proper generative models can also be used), which explicitly map data $\mathbf{x}$ with the encoder $q_\phi(\mathbf{z}|\mathbf{x}, \boldsymbol{\psi})$ to its corresponding latent code $\mathbf{z}$ and reconstruct the data from the latent code using the decoder $p_\theta(\mathbf{x}|\mathbf{z}, \boldsymbol{\psi})$. By maintaining this one-on-one mapping between the data and the latent code, cVAEs attempt to capture all modes of the data. As discussed in Sec. 3.1, cVAEs jointly optimize the encoder and decoder to maximize the variational lower bound:

$$\mathcal{L}(\mathbf{x}, \boldsymbol{\psi}; \theta, \phi) = \mathbb{E}_{q_\phi(\mathbf{z}|\mathbf{x}, \boldsymbol{\psi})} \left[ \log p_\theta(\mathbf{x}|\mathbf{z}, \boldsymbol{\psi}) \right] - \mathrm{KL}\left( q_\phi(\mathbf{z}|\mathbf{x}, \boldsymbol{\psi}) \| p(\mathbf{z}) \right). \tag{6}$$

We use multivariate Gaussians for the prior, encoder and decoder: $p(\mathbf{z}) = \mathcal{N}(\mathbf{z}; \mathbf{0}, \mathbf{I})$, $q_\phi(\mathbf{z}|\mathbf{x}, \boldsymbol{\psi}) = \mathcal{N}(\mathbf{z}; \boldsymbol{\mu}, \boldsymbol{\sigma}^2 \mathbf{I})$, and $p_\theta(\mathbf{x}|\mathbf{z}, \boldsymbol{\psi}) = \mathcal{N}(\mathbf{x}; \tilde{\mathbf{x}}, \alpha \mathbf{I})$. Both the encoder and decoder are implemented as neural networks. The encoder network $f_\phi$ outputs the parameters of the posterior distribution: $(\boldsymbol{\mu}, \boldsymbol{\sigma}) = f_\phi(\mathbf{x}, \boldsymbol{\psi})$. The decoder network $g_\theta$ outputs the reconstructed future trajectory $\tilde{\mathbf{x}}$:

$\tilde{\mathbf{x}} = g_\theta(\mathbf{z}, \boldsymbol{\psi})$. Detailed network architectures are given in Appendix B.1. Based on the Gaussian parameterization of the cVAE, the objective in Eq. 6 can be rewritten as

$$\mathcal{L}_{cvae}(\mathbf{x}, \boldsymbol{\psi}; \theta, \phi) = -\frac{1}{V}\sum_{v=1}^{V}\|\tilde{\mathbf{x}}_v - \mathbf{x}\|^2 + \beta \cdot \frac{1}{D_z}\sum_{j=1}^{D_z}\left(1 + 2\log\sigma_j - \mu_j^2 - \sigma_j^2\right), \qquad (7)$$

where we take $V$ samples from the posterior $q_\phi(\mathbf{z}|\mathbf{x}, \boldsymbol{\psi})$, $D_z$ is the number of latent dimensions, and $\beta = 1/\alpha$ is a weighting factor. The training procedure for the cVAE is detailed in Alg. 2 (Appendix A). Once the cVAE model is trained, sampling from the learned future trajectory model $p_\theta(\mathbf{x}|\boldsymbol{\psi})$ is efficient: we can sample a latent code $\mathbf{z}$ according to the prior $p(\mathbf{z})$ and use the decoder $p_\theta(\mathbf{x}|\mathbf{z}, \boldsymbol{\psi})$ to decode it into a future trajectory $\mathbf{x}$.

---

**Algorithm 1** Training the diversity sampling function (DSF) $\mathcal{S}_\gamma(\boldsymbol{\psi})$

---

1: **Input:** Training data $\{\mathbf{x}^{(i)}, \boldsymbol{\psi}^{(i)}\}_{i=1}^M$, cVAE decoder network $g_\theta(\mathbf{z}, \boldsymbol{\psi})$
2: **Output:** DSF $\mathcal{S}_\gamma(\boldsymbol{\psi})$
3: Initialize $\gamma$ randomly
4: **while** not converged **do**
5:     **for** each $\boldsymbol{\psi}^{(i)}$ **do**
6:         Generate latent codes $\mathcal{Z} = \{\mathbf{z}_1, \ldots, \mathbf{z}_N\}$ with the DSF $\mathcal{S}_\gamma(\boldsymbol{\psi})$
7:         Generate the trajectory ground set $\mathcal{Y} = \{\mathbf{x}_1, \ldots, \mathbf{x}_N\}$ with the decoder $g_\theta(\mathbf{z}, \boldsymbol{\psi})$
8:         Compute the similarity matrix $\mathbf{S}$ and quality vector $\mathbf{r}$ with Eq. 8 and 9
9:         Compute the DPP kernel $\mathbf{L}(\gamma) = \text{Diag}(\mathbf{r}) \cdot \mathbf{S} \cdot \text{Diag}(\mathbf{r})$
10:        Calculate the diversity loss $\mathcal{L}_{diverse}$
11:        Update $\gamma$ with the gradient $\nabla\mathcal{L}_{diverse}$
12:     **end for**
13: **end while**

---

## 4.2 DIVERSITY SAMPLING FUNCTION (DSF)

As mentioned before, randomly sampling from the learned cVAE model according to the implicit likelihood function $p_\theta(\mathbf{x}|\boldsymbol{\psi})$, *i.e.,* sampling latent codes from the prior $p(\mathbf{z})$, does not guarantee that the trajectory samples are diverse: major modes (those having more data) with higher likelihood will produce most of the samples while minor modes with lower likelihood will have almost no sample. This prompts us to devise a new sampling strategy that can reliably generate a diverse set of samples covering both major and minor modes. We propose to learn a diversity sampling function (DSF) $\mathcal{S}_\gamma(\boldsymbol{\psi})$ that maps context $\boldsymbol{\psi}$ to a set of latent codes $\mathcal{Z} = \{\mathbf{z}_1, \ldots, \mathbf{z}_N\}$. The DSF is implemented as a $\gamma$-parameterized neural network which takes $\boldsymbol{\psi}$ as input and outputs a vector of length $N \cdot D_z$ (see Appendix B.1 for network details). The latent codes $\mathcal{Z}$ are decoded into a diverse set of future trajectories $\mathcal{Y} = \{\mathbf{x}_1, \ldots, \mathbf{x}_N\}$, which are denoted as the DSF trajectory samples. We note that $N$ is the sampling budget. To solve for the parameters of the DSF, we propose a diversity loss based on a DPP defined over $\mathcal{Y}$. In this section, we first describe how the DPP kernel $\mathbf{L}$ is defined, which involves the construction of the similarity matrix $\mathbf{S}$ and quality vector $\mathbf{r}$. We then discuss how we use the DPP kernel $\mathbf{L}$ to formulate a diversity loss to optimize the parameters of the DSF.

Recall that the DPP kernel is defined as $\mathbf{L} = \text{Diag}(\mathbf{r}) \cdot \mathbf{S} \cdot \text{Diag}(\mathbf{r})$, where $\mathbf{r}$ defines the quality of each trajectory and $\mathbf{S}$ measures the similarity between two trajectories. The DPP kernel $\mathbf{L}(\gamma)$ is a function of $\gamma$ as it is defined over the ground set $\mathcal{Y}$ output by the DSF $\mathcal{S}_\gamma(\boldsymbol{\psi})$.

**Similarity.** We measure the similarity $\mathbf{S}_{ij}$ between two trajectories $\mathbf{x}_i$ and $\mathbf{x}_j$ as

$$\mathbf{S}_{ij} = \exp\left(-k \cdot d_\mathbf{x}^2(\mathbf{x}_i, \mathbf{x}_j)\right), \qquad (8)$$

where $d_\mathbf{x}$ is the Euclidean distance and $k$ is a scaling factor. This similarity design ensures that $0 \leq \mathbf{S}_{ij} \leq 1$ and $\mathbf{S}_{ii} = 1$. It also makes $\mathbf{S}$ positive definite since the Gaussian kernel we use is a positive definite kernel.

**Quality.** It may be tempting to use $p(\mathbf{x}|\boldsymbol{\psi})$ to define the quality of each trajectory sample. However, this likelihood-based measure will clearly favor major modes that have higher probabilities, making it less likely to generate samples from minor modes. This motivates us to design a quality metric that

treats all modes equally. To this end, unlike the similarity metric which is defined in the trajectory space, the quality of each sample is measured in the latent space and is defined as

$$
r_i = \begin{cases} \omega, & \text{if } \|\mathbf{z}_i\| \leq R \\ \omega \exp\left(-\mathbf{z}_i^T \mathbf{z}_i + R^2\right), & \text{otherwise} \end{cases} \tag{9}
$$

Geometrically, let $R$ be the radius of a sphere $\Phi$ containing most samples from the Gaussian prior $p(\mathbf{z})$. We treat samples inside $\Phi$ equally and only penalize samples outside $\Phi$. In this way, samples from major modes are not preferred over those from minor modes as long as they are inside $\Phi$, while samples far away from the data manifold are heavily penalized as they are outside $\Phi$. The radius $R$ is determined by where $\rho$ percent of the Gaussian samples lie within, and we set $\rho = 90$. To compute $R$, we use the percentage point function of the chi-squared distribution which models the distribution over the sum of squares of independent standard normal variables. The base quality $\omega$ is a hyperparameter which we set to 1 during training in our experiments. At test time, we can use a larger $\omega$ to encourage the DPP to select more items from the ground set $\mathcal{Y}$. The hyperparameter $\rho$ (or $R$) allows for the trade-off between diversity and quality. When $R$ is small, the quality metric is reduced to a pure likelihood-based metric (proportional to the latent likelihood), which will prefer samples with high likelihood and result in a less diverse sample set. When $R$ is large, most samples will have the same quality, and the resulting samples will be highly diverse but less likely. In practice, the choice of $R$ should be application dependent, as one could imagine autonomous vehicles would need to consider more diverse scenarios including those less likely ones to ensure robustness. We note that after the diverse samples are obtained, it is possible to reassign the quality score for each sample based on its likelihood to allow users to prioritize more likely samples.

**Diversity Loss.** To optimize the DSF $\mathcal{S}_\gamma(\boldsymbol{\psi})$, we need to define a diversity loss that measures the diversity of the trajectory ground set $\mathcal{Y}$ generated by $\mathcal{S}_\gamma(\boldsymbol{\psi})$. An obvious choice for the diversity loss would be the negative log likelihood $-\log \mathcal{P}_{\mathbf{L}(\gamma)}(\mathbf{Y} = \mathcal{Y}) = -\log\det(\mathbf{L}(\gamma)) + \log\det(\mathbf{L}(\gamma) + \mathbf{I})$. However, there is a problem with directly using the DPP log likelihood. The log likelihood heavily penalizes repeated items: if two trajectories inside $\mathcal{Y}$ are very similar, their corresponding rows in $\mathbf{L}$ will be almost identical, making $\det(\mathbf{L}(\gamma)) = \lambda_1 \lambda_2 \ldots \lambda_N \approx 0$ ($\lambda_n$ is the $n$-th eigenvalue). In practice, if the number of modes in the trajectory distribution $p(\mathbf{x}|\boldsymbol{\psi})$ is smaller than $|\mathcal{Y}|$, $\mathcal{Y}$ will always have similar trajectories, thus making $\det(\mathbf{L}(\gamma))$ always close to zero. In such cases, optimizing the negative log likelihood causes numerical issues, which is observed in our early experiments.

Instead, the expected cardinality of the DPP is a better measure for the diversity of $\mathcal{Y}$, which is defined as $\mathbb{E}_{\mathbf{Y} \sim \mathcal{P}_{\mathbf{L}(\gamma)}}[|\mathbf{Y}|]$. Intuitively, since the DPP discourages selection of similar items, if $\mathcal{Y}$ is more diverse, a random subset $\mathbf{Y}$ drawn according to the DPP is more likely to select more items from $\mathcal{Y}$, thus having larger cardinality. The expected cardinality can be computed as (Eq. 15 and 34 in Kulesza et al. (2012)):

$$
\mathbb{E}[|\mathbf{Y}|] = \sum_{n=1}^{N} \frac{\lambda_n}{\lambda_n + 1} = \text{tr}\left(\mathbf{I} - (\mathbf{L}(\gamma) + \mathbf{I})^{-1}\right). \tag{10}
$$

The main advantage of the expected cardinality is that it is well defined even when the ground set $\mathcal{Y}$ has duplicated items, since it does not require all eigenvalues of $\mathbf{L}$ to be non-zero as the log likelihood does. Thus, our diversity loss is defined as $\mathcal{L}_{diverse}(\gamma) = -\text{tr}\left(\mathbf{I} - (\mathbf{L}(\gamma) + \mathbf{I})^{-1}\right)$. The training procedure for $\mathcal{S}_\gamma(\boldsymbol{\psi})$ is outlined in Alg. 1.

**Inference.** At test time, given current context $\boldsymbol{\psi}$, we use the learned DSF $\mathcal{S}_\gamma(\boldsymbol{\psi})$ to generate the future trajectory ground set $\mathcal{Y}$. In some cases, $\mathcal{Y}$ may still contain some trajectories that are similar to others. In order to obtain a diverse set of trajectories without repetition, we aim to perform MAP inference for the DPP to find the most diverse subset $Y^* = \arg\max_{Y \in \mathcal{Y}} \mathcal{P}_{\mathbf{L}(\gamma)}(Y)$. A useful property of DPPs is that the log-probability function is submodular (Gillenwater et al., 2012). Even though submodular maximization is NP-hard, we use a greedy algorithm (Nemhauser et al., 1978) which is a popular optimization procedure that works well in practice. As outlined in Alg. 3, the output set $Y_f$ is initialized as $\emptyset$, and at each iteration, the trajectory which maximizes the log probability

$$
\mathbf{x}^* = \underset{\mathbf{x} \in \mathcal{Y} \backslash Y_f}{\arg\max} \ \log\det\left(\mathbf{L}_{Y_f \cup \{\mathbf{x}\}}\right) \tag{11}
$$

is added to $Y_f$, until the marginal gain becomes negative or $Y_f = \mathcal{Y}$.

## 5 EXPERIMENTS

The primary focus of our experiments is to answer the following questions: (1) Are trajectory samples generated with our diversity sampling function more diverse than samples from the cVAE and other baselines? (2) How does our method perform on both balanced and imbalanced data? (3) Is our method general enough to perform well on both low-dimensional and high-dimensional tasks?

**Metrics.** A problem with trajectory forecasting evaluation is that in real data each context $\boldsymbol{\psi}^{(i)}$ usually only has one future trajectory $\mathbf{x}^{(i)}$, which means we only have one sample from a multi-modal distribution. Let us consider a scenario of three data examples $\{\mathbf{x}^{(i)}, \boldsymbol{\psi}^{(i)}\}_{i=1}^3$ as shown in Fig. 2 (red, purple, blue). The contexts (past trajectories) of the three examples are instances of the same trajectory but they are slightly different due to noise. As these three contexts have the same semantic meaning, they should share the future trajectories, *e.g.*, the purple and blue future trajectories are also valid for the red context. If we evaluate each example $(\mathbf{x}^{(i)}, \boldsymbol{\psi}^{(i)})$ only with its own future trajectory $\mathbf{x}^{(i)}$, a method

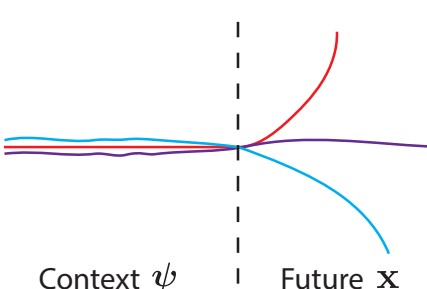

Figure 2: In real data, contexts (past trajectories) are seldom the same due to noise.

can achieve high scores by only forecasting the mode corresponding to $\mathbf{x}^{(i)}$ and dropping other modes. This is undesirable because we want a good method to capture all modes of the future trajectory distribution, not just a single mode. To allow for multi-modal evaluation, we propose collecting multiple future trajectories for each example by clustering examples with similar contexts. Specifically, we augment each data example $(\mathbf{x}^{(i)}, \boldsymbol{\psi}^{(i)})$ with a future trajectory set $\mathcal{X}^{(i)} = \{\mathbf{x}^{(j)} | \|\boldsymbol{\psi}^{(j)} - \boldsymbol{\psi}^{(i)}\| \leq \varepsilon, j = 1, \ldots, M\}$ and metrics are calculated based on $\mathcal{X}^{(i)}$ instead of $\mathbf{x}^{(i)}$, *i.e.,* we compute metrics for each $\mathbf{x} \in \mathcal{X}^{(i)}$ and average the results.

We use the following metrics for evaluation: (1) **Average Displacement Error (ADE)**: average mean square error (MSE) over all time steps between the ground truth future trajectory $\mathbf{x}$ and the closest sample $\tilde{\mathbf{x}}$ in the forecasted set of trajectories $Y_f$. (2) **Final Displacement Error (FDE)**: MSE between the final ground truth position $\mathbf{x}^T$ and the closest sample's final position $\tilde{\mathbf{x}}^T$. (3) **Average Self Distance (ASD)**: average $L2$ distance over all time steps between a forecasted sample $\tilde{\mathbf{x}}_i$ and its closest neighbor $\tilde{\mathbf{x}}_j$ in $Y_f$. (4) **Final Self Distance (FSD)**: $L2$ distance between the final position of a sample $\tilde{\mathbf{x}}_i^T$ and its closest neighbor's final position $\tilde{\mathbf{x}}_j^T$. The ADE and FDE are common metrics used in prior work on trajectory forecasting (Alahi et al., 2016; Lee et al., 2017; Rhinehart et al., 2018; Gupta et al., 2018). However, these two metrics do not penalize repeated samples. To address this, we introduce two new metrics ASD and FSD to evaluate the similarity between samples in the set of forecasted trajectories. Larger ASD and FSD means the forecasted trajectories are more non-repetitive and diverse.

**Baselines.** We compare our **Diversity Sampler Function (DSF)** with the following baselines: (1) **cVAE**: a method that follows the original sampling scheme of cVAE by sampling latent codes from a Gaussian prior $p(\mathbf{z})$. (2) **MCL**: an approach that uses multiple choice learning (Lee et al., 2016) to optimize the sampler $\mathcal{S}_\gamma(\boldsymbol{\psi})$ with the following loss: $\mathcal{L}_{\text{mcl}} = \min_{\tilde{\mathbf{x}} \in \mathcal{Y}} \|\tilde{\mathbf{x}} - \mathbf{x}\|^2$, where $\mathbf{x}$ is the ground truth future trajectory. (3) **R2P2**: a method proposed in (Rhinehart et al., 2018) that uses a reparametrized pushforward policy to improve modeling of multi-modal distributions for vehicle trajectories. (4) **cGAN**: generative adversarial networks (Goodfellow et al., 2014) conditioned on the forecasting context. We implement all baselines using similar networks and perform hyperparameter search for each method for fair comparisons. For methods whose samples are stochastic, we use 10 random seeds and report the average results for all metrics.

### 5.1 SYNTHETIC 2D TRAJECTORY DATA

We first use synthetic data to evaluate our method's performance for low-dimensional tasks. We design a virtual 2D traffic scene where a vehicle comes to a crossroad and can choose three different future routes: forward, left, and right. We consider two types of synthetic data: (1) Balanced data, which means the probabilities of the vehicle choosing one of the three routes are the same; (2)

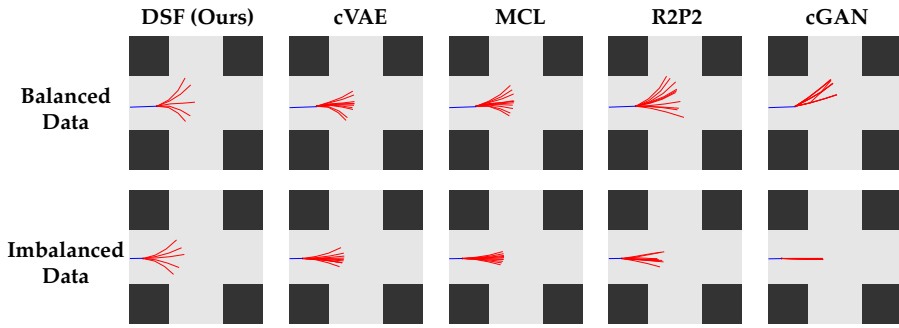

Figure 3: Qualitative results on synthetic data for both balanced and imbalanced data distribution when $N = 10$. Blue represents the past trajectory and red represents forecasted future trajectories.

| | Balanced Data | | | | Imbalanced Data | | | |
|---|---|---|---|---|---|---|---|---|
| Method | ADE ↓ | FDE ↓ | ASD ↑ | FSD ↑ | ADE ↓ | FDE ↓ | ASD ↑ | FSD ↑ |
| DSF (Ours) | **0.182** | **0.344** | **0.147** | **0.340** | **0.198** | **0.371** | **0.207** | **0.470** |
| cVAE | 0.262 | 0.518 | 0.022 | 0.050 | 0.332 | 0.662 | 0.021 | 0.050 |
| MCL | 0.276 | 0.548 | 0.012 | 0.030 | 0.457 | 0.938 | 0.005 | 0.010 |
| R2P2 | 0.211 | 0.361 | 0.047 | 0.080 | 0.393 | 0.776 | 0.019 | 0.030 |
| cGAN | 0.808 | 1.619 | 0.018 | 0.010 | 1.784 | 3.744 | 0.006 | 0.001 |

Table 1: Quantitative results on synthetic data (numbers scaled by 10) when $N = 10$.

Imbalanced data, where the probabilities of the vehicle going forward, left and right are 0.8, 0.1, 0.1, respectively. We synthesize trajectory data by simulating the vehicle's behavior and adding Gaussian noise to vehicle velocities. Each data example $(\mathbf{x}^{(i)}, \boldsymbol{\psi}^{(i)})$ contains future trajectories of 3 steps and past trajectories of 2 steps. We also add an obstacle map around the current position to the context $\boldsymbol{\psi}^{(i)}$. In total, we have around 1100 training examples and 1000 test examples. Please refer to Appendix B for more implementation details.

Table 1 summarizes the quantitative results for both balanced and imbalanced data when the sampling budget $N$ is 10. We can see that our method DSF outperforms the baselines in all metrics in both test settings. As shown in Fig. 3, our method generates more diverse trajectories and is less affected by the imbalanced data distribution. The trajectory samples of our method are also less repetitive, a feature afforded by our DPP formulation. Fig. 4 shows how ADE changes as a function of the sampling budget $N$.

## 5.2 DIVERSE HUMAN MOTION FORECASTING

To further evaluate our method's performance for more complex and high-dimensional tasks, we apply our method to forecast future human motions (pose sequences). We use motion capture to obtain 10 motion sequences including different types of motions such as walking, turning, jogging, bending, and crouching. Each sequence is about 1 minute long and each pose consists of 59 joint angles. We use past 3 poses

| Method | ADE ↓ | FDE ↓ | ASD ↑ | FSD ↑ |
|---|---|---|---|---|
| DSF (Ours) | **0.259** | **0.421** | **0.115** | **0.282** |
| cVAE | 0.332 | 0.642 | 0.034 | 0.098 |
| MCL | 0.344 | 0.674 | 0.036 | 0.122 |
| cGAN | 0.652 | 1.296 | 0.001 | 0.003 |

Table 2: Quantitative results on for human motion forecasting when $N = 10$.

(0.1s) to forecast next 30 poses (1s). There are around 9400 training examples and 2000 test examples where we use different sequences for training and testing. More implementation details can be found in Appendix B.

We present quantitative results in Table 2 and we can see that our approach outperforms other methods in all metrics. As the dynamics model used in R2P2 (Rhinehart et al., 2018) does not generalize well for high-dimensional human motion, we find the model fails to converge and we do not compare with it in this experiment. Fig. 4 shows that our method achieves large improvement when the sampling budget is big ($N = 50$). We also present qualitative results in Fig. 5, where we show the starting pose and the final pose of all 10 forecasted motion samples for each method. We can clearly

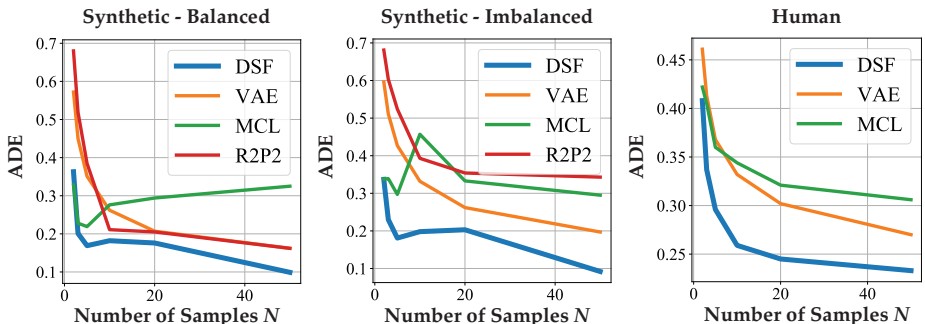

Figure 4: ADE vs. $N$ for synthetic data and human motion forecasting. cGAN is not shown in this plot as it is much worse than other methods due to mode collapse.

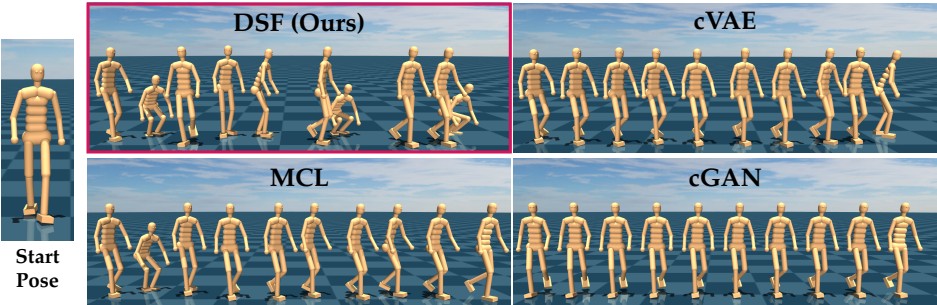

Figure 5: Qualitative results for human motion forecasting when $N = 10$. The left shows the starting pose, and the right shows for each method the final pose of all 10 forecasted motion samples.

see that our method generates more diverse future human motions than the baselines. Please refer to Appendix C and our video for additional qualitative results.

### 5.3 ADDITIONAL EXPERIMENTS WITH DIVERSITY-BASED BASELINES

In this section, we perform additional experiments on a large human motion dataset (3.6 million frames), Human3.6M (Ionescu et al., 2013), to evaluate the generalization ability of our approach. We predict future motion of 2 seconds based on observed motion of 0.5 seconds. Please refer to Appendix B.3 for implementation details. We also use a new selection of baselines including several variants of our method (DSF) and the cVAE to validate several design choices of our method, including the choice of the expected cardinality over the negative log likelihood (NLL) of the DPP as the diversity loss. Specifically, we use the following new baselines: (1) **DSF-NLL**: a variant of DSF that uses NLL as the diversity loss instead of the expected cardinality. (2) **DSF-COS**: a DSF variant that uses cosine similarity to build the similarity matrix $\mathbf{S}$ for the DPP kernel $\mathbf{L}$. (3) **DSF-NLL**: a variant of the cVAE that samples 100 latent codes and performs DPP MAP inference on the latent codes to obtain a diverse set of latent codes, which are then decoded into trajectory samples.

We present quantitative results in Table 3 when the number of samples $N$ is 10 and 50. The baseline DSF-COS is able to achieve very high diversity (ASD and FSD) but its samples are overly diverse and have poor quality which is indicated by the large ADE and FDE. Compared with DSF-NLL,

|        | $N = 10$ | | | | $N = 50$ | | | |
| Method | ADE ↓ | FDE ↓ | ASD ↑ | FSD ↑ | ADE ↓ | FDE ↓ | ASD ↑ | FSD ↑ |
| --- | --- | --- | --- | --- | --- | --- | --- | --- |
| DSF (Ours) | **0.340** | **0.521** | 0.381 | 0.621 | **0.236** | **0.306** | 0.313 | 0.415 |
| DSF-NLL | **0.335** | **0.514** | 0.343 | 0.496 | X | X | X | X |
| DSF-COS | 2.588 | 1.584 | **5.093** | **5.718** | 0.978 | 0.891 | **2.007** | **1.968** |
| cVAE | 0.363 | 0.549 | 0.235 | 0.360 | 0.276 | 0.369 | 0.160 | 0.220 |
| cVAE-LDPP | 0.373 | 0.554 | 0.280 | 0.426 | 0.277 | 0.365 | 0.176 | 0.240 |

Table 3: Quantitative results on Human3.6M (Ionescu et al., 2013) for $N = 10$ and $N = 50$. X means the method is unable to learn a model due to numerical instability.

our method achieves better diversity (ASD and FSD) and similar ADE and FDE when the number of samples is small ($N = 10$). For a larger number of samples ($N = 50$), NLL becomes unstable even with a large $\epsilon$ (1e-3) added to the diagonal. This behavior of NLL, *i.e.,* stable for small $N$ but unstable for large $N$, matches our intuition that NLL becomes unstable when samples become similar (as discussed in Sec. 4.2), because when there are more samples, it is easier to have similar samples during the SGD updates of the DSF network. The baseline cVAE-LDPP also performs worse than DSF in all metrics even though it is able to outperfom the cVAE. We believe the reason is that diversity in sample space may not be well reflected in the latent space due to the non-linear mapping from latent codes to samples induced by deep neural networks.

## 6 CONCLUSION

We proposed a novel forecasting approach using a DSF to optimize over the sample space of a generative model. Our method learns the DSF with a DPP-based diversity measure to generate a diverse set of trajectories. The diversity measure is a novel application of DPPs to optimize a set of items in continuous space. Experiments have shown that our approach can generate more diverse vehicle trajectories and human motions compared to state-of-the-art baseline forecasting approaches.

**Acknowledgment**. This project was sponsored in part by JST CREST (JPMJCR14E1), NSF NRI (1637927) and IARPA (D17PC00340).

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

# A   ALGORITHMS

---

**Algorithm 2** Training the cVAE

---

1: **Input:** Training data $\{\mathbf{x}^{(i)}, \boldsymbol{\psi}^{(i)}\}_{i=1}^M$
2: **Output:** cVAE encoder network $f_\phi(\mathbf{x}, \boldsymbol{\psi})$ and decoder network $g_\theta(\mathbf{z}, \boldsymbol{\psi})$
3: Initialize $\phi$ and $\theta$ randomly
4: **while** not converged **do**
5:     **for** each $(\mathbf{x}^{(i)}, \boldsymbol{\psi}^{(i)})$ **do**
6:         Compute parameters $(\boldsymbol{\mu}, \boldsymbol{\sigma})$ of the posterior distribution $q_\phi(\mathbf{z}|\mathbf{x}, \boldsymbol{\psi})$ using $f_\phi(\mathbf{x}, \boldsymbol{\psi})$
7:         Sample $V$ Gaussian noises $\{\boldsymbol{\epsilon}_1, \ldots, \boldsymbol{\epsilon}_V\}$ from $\mathcal{N}(\mathbf{0}, \mathbf{I})$
8:         Transform noises to latent samples from $q_\phi(\mathbf{z}|\mathbf{x}, \boldsymbol{\psi})$: $\mathbf{z}_v = \boldsymbol{\mu} + \boldsymbol{\sigma} \odot \boldsymbol{\epsilon}_v$
9:         Decode latent samples into reconstructed trajectories $\{\tilde{\mathbf{x}}_1, \ldots, \tilde{\mathbf{x}}_V\}$ using $g_\theta(\mathbf{z}, \boldsymbol{\psi})$
10:         Calculate the cVAE loss $\mathcal{L}_{cvae}$ according to Eq. 6
11:         Update $\phi$ and $\theta$ with $\nabla_\phi \mathcal{L}_{cvae}$ and $\nabla_\theta \mathcal{L}_{cvae}$
12:     **end for**
13: **end while**

---

**Algorithm 3** Inference with the DSF $\mathcal{S}_\gamma(\boldsymbol{\psi})$

---

1: **Input:** Context $\boldsymbol{\psi}$, DSF $\mathcal{S}_\gamma(\boldsymbol{\psi})$, cVAE decoder network $g_\theta(\mathbf{z}, \boldsymbol{\psi})$
2: **Output:** Forecasted trajectory set $Y_f$
3: Generate latent codes $\mathcal{Z} = \{\mathbf{z}_1, \ldots, \mathbf{z}_N\}$ with the DSF $\mathcal{S}_\gamma(\boldsymbol{\psi})$
4: Generate the trajectory ground set $\mathcal{Y} = \{\mathbf{x}_1, \ldots, \mathbf{x}_N\}$ with the decoder $g_\theta(\mathbf{z}, \boldsymbol{\psi})$
5: Compute the DPP kernel $\mathbf{L} = \mathrm{Diag}(\mathbf{r}) \cdot \mathbf{S} \cdot \mathrm{Diag}(\mathbf{r})$
6: $Y_f \leftarrow \emptyset, U \leftarrow \mathcal{Y}$
7: **while** $U$ is not empty **do**
8:     $\mathbf{x}^* \leftarrow \arg\max_{\mathbf{x} \in U} \log\det\left(\mathbf{L}_{Y_f \cup \{\mathbf{x}\}}\right)$
9:     **if** $\log\det\left(\mathbf{L}_{Y_f \cup \{\mathbf{x}^*\}}\right) - \log\det\left(\mathbf{L}_{Y_f}\right) < 0$ **then**
10:         **break**
11:     **end if**
12:     $Y_f \leftarrow Y_f \cup \{\mathbf{x}^*\}$
13:     $U \leftarrow U \setminus \{\mathbf{x}^*\}$
14: **end while**

---

# B  IMPLEMENTATION DETAILS

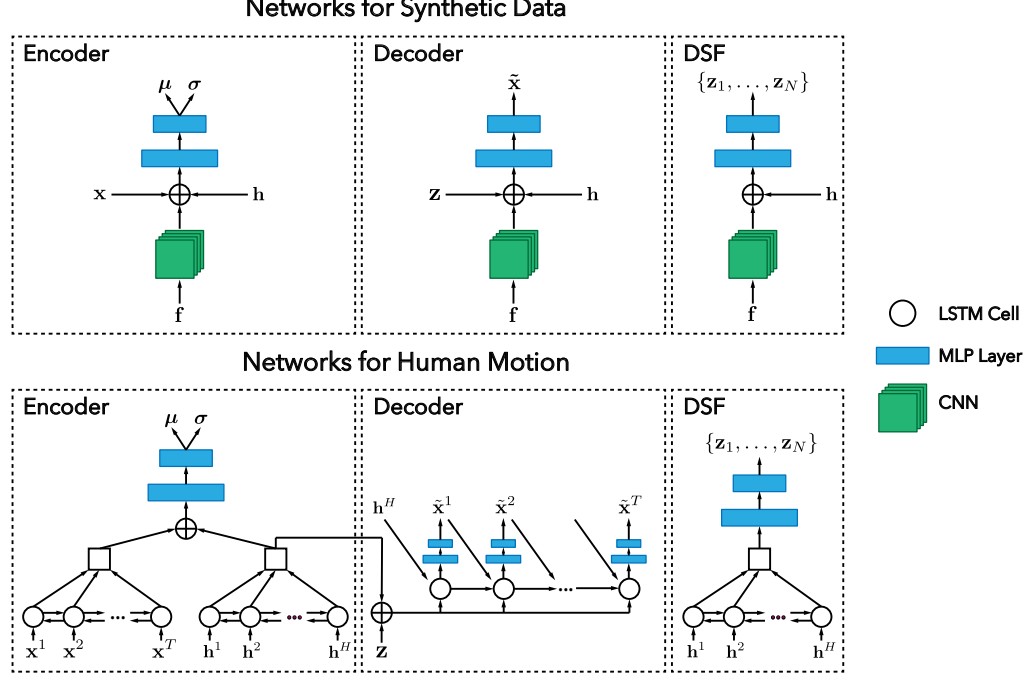

Figure 6: Network architectures for synthetic data and human motion. **Top:** for synthetic data, we use a CNN to process the obstacle map $\mathbf{f}$ and directly flatten trajectories $\mathbf{x}$ and $\mathbf{h}$ into vectors. The reconstructed trajectory $\tilde{\mathbf{x}}$ is decoded with an MLP. **Bottom:** for human motion, we use Bi-LSTMs to extract temporal features for $\mathbf{x}$ and $\mathbf{h}$ and decode the reconstructed trajectory $\tilde{\mathbf{x}}$ with a forward LSTM.

## B.1  NETWORK ARCHITECTURES

**Synthetic data.** Fig. 6 (Top) shows the network architecture for synthetic data. The number of latent dimensions is 2. By default, we use ReLU activation for all networks. The future trajectory $\mathbf{x} \in \mathbb{R}^{3 \times 2}$ consists of 3 future positions of the vehicle. The context $\psi$ contains past trajectories $\mathbf{h} \in \mathbb{R}^{2 \times 2}$ of 2 time steps and a obstacle map $\mathbf{f} \in \{0, 1\}^{28 \times 28}$ spanning a $4 \times 4$ area around the current position of the vehicle (the road width is 2). For the encoder, we use a convolutional neural network (CNN) with three 32-channel convolutional layers to process $\mathbf{f}$. The first two layers have kernel size 4 and stride 2 while the last layer has kernel size 6 and stride 1. The obtained CNN features are concatenated with flattened $\mathbf{x}$ and $\mathbf{h}$ into a unified feature, which is feed into a multilayer perceptron (MLP). The MLP has one 128-dim hidden layer and two heads outputing the mean $\boldsymbol{\mu}$ and variance $\boldsymbol{\sigma}$ of the latent distribution. For the decoder, we concatenate the CNN feature from $\mathbf{f}$ with the latent code $\mathbf{z} \in \mathbb{R}^2$ and flattened $\mathbf{h}$ into a unified feature. The feature is passed through an MLP with one 128-dim hidden layer which outputs the reconstructed future trajectory $\tilde{\mathbf{x}} \in \mathbb{R}^{3 \times 2}$. For the diversity sampler function (DSF), we concatenate the CNN feature from $\mathbf{f}$ with the flattened $\mathbf{h}$ and pass it through an MLP with one 128-dim hidden layer to obtain a set of latent codes $\{\mathbf{z}_1, \ldots, \mathbf{z}_N\}$ which are represented by a vector of length $2N$.

**Human motion.** Fig. 6 (Bottom) shows the network architecture for synthetic data. The number of latent dimensions is 8. The future trajectory $\mathbf{x} \in \mathbb{R}^{30 \times 59}$ consists of future poses of 30 time steps (1s). The context $\psi$ contains past poses $\mathbf{h} \in \mathbb{R}^{3 \times 59}$ of 3 time steps (0.1s). Each pose consists of 59 joint angles. For the encoder, we use two 128-dim bidirectional LSTMs (Bi-LSTMs) and mean pooling to obtain the temporal features for $\mathbf{x}$ and $\mathbf{h}$. We then concatenate the temporal features into a unified feature and feed it into an MLP with two hidden layers $(300, 200)$ and two heads to obtain the mean $\boldsymbol{\mu}$ and variance $\boldsymbol{\sigma}$ of the latent distribution. For the decoder, we reuse the Bi-LSTM of

the encoder for the context $\mathbf{h}$ and a 128-dim forward LSTM to decode the future trajectory $\tilde{\mathbf{x}}$. At each time step $t$, the forward LSTM takes as input the previous pose $\tilde{\mathbf{x}}^{t-1}$ ($\mathbf{h}^H$ for $t = 0$), the latent code $\mathbf{z} \in \mathbb{R}^8$ and the temporal features from $\mathbf{h}$, and outputs a 128-dim feature. The feature is then passed through an MLP with two hidden layers $(300, 200)$ to generate the reconstructed pose $\tilde{\mathbf{x}}^t$. For the DSF, we use a different 128-dim Bi-LSTM to obtain the temporal feature for $\mathbf{h}$, which is feed into an MLP with a 128-dim hidden layer to produce a set of latent codes $\{\mathbf{z}_1, \ldots, \mathbf{z}_N\}$ which are represented by a vector of length $8N$.

## B.2   TRAINING AND EVALUATION

When training the cVAE model using Eq. 7, we take $V = 1$ sample from the posterior $q_\phi(\mathbf{z}|\mathbf{x}, \boldsymbol{\psi})$. The weighting factor $\beta$ for the KL term is set to 0.1 for synthetic data and 1e-4 for human motion. We use Adam (Kingma and Ba, 2014) to jointly optimize the encoder and decoder. The learning rate is set to 1e-4 and we use a mini batch size of 32 for synthetic data. We optimize the model for 500 epochs for synthetic data and 100 epochs for human motion.

When training the DSF, the scale factor $k$ for the similarity matrix $\mathbf{S}$ is set to 1 for synthetic data and 1e-2 for human motions. For both synthetic data and human motions, we use Adam with learning rate 1e-4 to optimize the DSF for 20 epochs.

Recall that in the metrics section (Sec. 5.1), we need the grouping threshold $\varepsilon$ to build the ground truth future trajectory set $\mathcal{X}^{(i)} = \{\mathbf{x}^{(j)} | \|\boldsymbol{\psi}^{(j)} - \boldsymbol{\psi}^{(i)}\| \leq \varepsilon,\ j = 1, \ldots, M\}$. For synthetic data, $\varepsilon$ is set to 0.1 and we only use past trajectories $\mathbf{h}$ to compute the distance between contexts. For human motion, $\varepsilon$ is set to 0.5.

## B.3   IMPLEMENTATION DETAILS FOR EXPERIMENTS ON HUMAN3.6M

Following previous work (Martinez et al., 2017; Pavlakos et al., 2017; Pavllo et al., 2019), we convert the motion sequences in the dataset into sequences of 3D joint positions, and adopt a 17-joint skeleton. We train on five subjects (S1, S5, S6, S7, S8), and test on two subjects (S9 and S11).

We use the same network architectures (Fig.6 (Bottom)) in this experiment as the one used in the human motion forecasting experiment above. The number of latent dimensions is 128. When training the cVAE model, the weighting factor $\beta$ is set to 0.1. We sample 5000 training examples every epoch and optimize the cVAE for 500 epochs using Adam and a learning rate of 1e-4. We set the batch size to 64 for the optimization.

The scale factor $k$ for the similarity matrix $\mathbf{S}$ of the DPP kernel is set to 5. When learning the DSF, we use a batch size of 64 and sample 1000 training examples every epoch and optimize the DSF for 20 epochs using Adam and a learning rate of 1e-3.

When computing the metrics, we set the grouping threshold $\varepsilon$ to 0.1.

## C  ADDITIONAL VISUALIZATION

We also show additional qualitative results for human motion forecasting in Fig. 7. The quality and diversity of the forecasted motions are best seen in our video[2].

Figure 7: Additional visualization for human motion forecasting. The left shows the starting pose, and on the right we show for each method the final pose of 10 forecasted motion samples.

---

[2]https://youtu.be/5i71SU_IdS4

