# OpenReview forum: "Diverse Trajectory Forecasting with Determinantal Point Processes"
_ICLR.cc/2020/Conference — Accept (Poster)_

### Official Review · AnonReviewer3 · 2019-10-21
**Official Blind Review #3**

**Rating:** 6

**Review:**

Caveat: I am very familiar with DPPs, but unfamiliar with the literature of trajectory forecasting, autonomous driving, etc.

Summary: This work introduces a generative model for diverse sequences based on a DPP, with the goal of providing likely yet non-overlapping possible future trajectories to models that require such information for safety concerns (eg, autonomous driving). Rather than using the DPPs negative log-likelihood as a measure of the trajectory set's diversity, the authors use the DPP's expected sample size as a proxy for a diversity metric. Experimentally, the authors show on two tasks that this approach generates more diverse, relevant trajectories than several competing baselines.

Recommendation: I recommend this paper be accepted, with the caveat that I am not well equipped to evaluate the experimental contribution of this paper, which in my opinion is the most important part of this work. I would have liked to see more extensive experiments, in particular (a) other baselines (DPP NLL as a diversity loss, different DPP kernels) and (b) experiments on larger datasets.

High level comments and questions:

- When generating the trajectory set (Alg. 3), your algorithm may generate sets of variable size (as the DPP NLL is log-submodular but not necessarily increasing). Did you also consider generating sets of fixed size, or of a minimal size?

- Previous work has also looked at using DPPs to alleviate mode collapse in GANs (Elfeki et al., ICML 2019). Could you comment on how you expect such a method would perform on your chosen tasks?

- The intuition that the DPP negative log-likelihood will overwhelmingly penalize subsets that contain a few similar trajectories seems very reasonable. However, I would have liked to see that intuition verified through an experiment that used the NLL as a diversity loss.

- The training sets for both experiments seem fairly small (1100 and 9400 training examples). Could you comment on this choice, and on whether you expect your experimental results to generalize to much larger datasets?

- Your motivating example is autonomous vehicles and safety concerns; with that example in mind, could you comment on how to select the radius R (or, equivalently, \rho) in Eq. 9? It seems like this choice would have significant downstream implications on the trade-off between accuracy and robustness to unpredictability.

- The idea of using the expected subset size as a metric for diversity is compelling. The authors may want to take a look at (Gillenwater et al., NeurIPS'18), which uses a similar metric as a proxy for user engagement.

Minor comments:
- after Eq. 2: please consider defining the acronym ELBO before using it.
- Eq. 8: if you are using an exponentiated quadratic, I believe the distance should be squared.
- Eq. 10: consider citing the relevant proposition from (Kulesza & Taskar).


**Experience Assessment:**

I do not know much about this area.

**Review Assessment: Checking Correctness Of Derivations And Theory:**

I carefully checked the derivations and theory.

**Review Assessment: Checking Correctness Of Experiments:**

I assessed the sensibility of the experiments.

**Review Assessment: Thoroughness In Paper Reading:**

I read the paper at least twice and used my best judgement in assessing the paper.

---

> ### Author Response · Authors · 2019-11-08
> **Response to Reviewer #3 (Part 1)**
>
> Thank you for your review and feedback. In the following, we will address your comments and concerns, especially the ones w.r.t. experiments.
>
> ** Variable size of the output set **
>
> For now, we only consider generating sets of variable size instead of a fixed or minimal size, because we want to be able to filter out redundant samples for down-stream tasks. For example, if there is indeed only one possible trajectory and all N samples are the same, we want to just output that one trajectory and remove all duplicates.
>
>
> ** Comments on (Elfeki et al., ICML 2019) **
>
> (Elfeki et al., ICML 2019) is an interesting work with great results and we will cite it in a revised version.
> I would expect (Elfeki et al., ICML 2019) to be able to capture all modes of the data distribution as the cVAE does. But since the sampling procedure in the paper is still the same as traditional generative models, i.e., independently sampling latent codes according to the prior and no repulsion between samples is modeled, it should perform similarly to the cVAE which will generate most samples from major modes for imbalanced data. It would be interesting to combine our approach with theirs to further improve the sample diversity of GANs.
>
>
> ** NLL Baseline **
>
> We indeed compare with NLL in our new experiments below. In short, NLL is stable for a small number of samples (10), but for a larger number of samples (50) it still leads to numerical instability.
>
>
> ** Large dataset experiments and more baselines (Requested by R2 & R3) **
>
> Our method can generalize to larger datasets as shown in the new experiments below.
>
> Dataset: We use a large public human motion dataset *Human3.6M* to perform a new experiment on human motion forecasting. We use past 0.5s to forecast future motions in 2s.
>
> Baselines:
> (1) NLL (requested by R2 & R3):  using the negative log-likelihood of the ground set under DPP as the loss. We also add eps * I to improve its stability.
> (2) DSF-COS (different kernels as requested by R3):  using cosine similarity as the similarity measure for the DPP kernel.
> (3) cVAE: random sampling from the Gaussian prior.
> (4) cVAE-LDPP (requested by R2):  We sample 100 latent codes and use DPP MAP inference on the latent codes to obtain N diverse latent codes, which are decoded into a set of trajectories.
>
> Results:
> For the metrics, lower (ADE, FDE) and larger (ASD, FSD) are better.
>
>  Number of samples N = 10
>  ════════════════════════════
>                                 ADE     FDE     ASD     FSD
>  ────────────────────
>   DSF (Ours)        0.340   0.521   0.381   0.621
>   NLL                     0.335   0.514   0.343   0.496
>   DSF-COS            2.588   1.584   5.093   5.718
>   cVAE                   0.363   0.549   0.235   0.360
>   cVAE-LDPP        0.373   0.554   0.280   0.426
>  ════════════════════════════
>
>  Number of samples N = 50
>  ════════════════════════════
>                               ADE     FDE     ASD     FSD
>  ────────────────────
>   DSF (Ours)       0.236   0.306   0.313   0.415
>   NLL                          X          X          X           X
>   DSF-COS           0.978   0.891   2.007   1.968
>   cVAE                  0.276   0.369   0.160   0.220
>   cVAE-LDPP       0.277   0.365   0.176   0.240
>  ════════════════════════════
> *X means that NLL is unable to learn a model due to numerical instability
>
> (1) Comparison with a different kernel:
> The baseline DSF-COS uses cosine similarity for the DPP kernel, we can see that it is able to achieve high diversity but its samples are overly diverse and have poor quality which is indicated by large ADE and FDE.
>
> (2) DSF vs NLL:
> We can see that for a small number of samples (N=10), our method achieves better diversity (larger ASD, FSD) and similar ADE, FSD to NLL. For a larger number of samples (N=50), NLL becomes unstable even with a large eps=1e-3 added to the diagonal. This behavior of NLL, i.e., stable for small N but unstable for large N, matches our intuition that NLL becomes unstable when samples become similar because when there are more samples, it is easier to have similar trajectories. And the reason why NLL cannot handle similar trajectories is that the DPP kernel L becomes singular as discussed in the paper (Section 4.2).
>
> (3) DPP MAP inference in the latent space:
> For the cVAE-LDPP, it is indeed able to generate more diverse samples than cVAE but the diversity is worse than DSF. We think the main reason is that the mapping from the latent space to the data space is highly non-linear, and a diverse set of latent codes won’t necessarily result in a diverse set of samples.

---

> > ### Author Response · Authors · 2019-11-08
> > **Response to Reviewer #3 (Part 2)**
> >
> >
> > ** Selecting radius R and $\rho$ **
> >
> > The trade-off between accuracy (quality) and robustness (diversity) should be application dependent. For example, safety-critical applications should use a larger R to consider a more diverse set of trajectory samples. We can also assign a score to each sample based on the latent code likelihood so that minor modes will have lower scores, this will allow users to rank the trajectories and prioritize more probable trajectories.
> >
> > Moreover, it would be interesting to condition the DSF on these user-specific parameters (R, k), so that users can achieve the trade-off between robustness and quality dynamically without the need for retraining the DSF.
> >
> >
> > ** Comparison with MIC **
> >
> > We will definitely cite [1] in a revised version as it is a great paper and is closely related. We agree that [1] and we both use the cardinality of a DPP as a metric, but we want to point out two important differences:
> > (1) The context is different. [1] is focused on the subset selection problem, so it is called maximum *induced* cardinality (MIC). If we understand correctly, the theories and approximation methods developed in [1] are geared toward subset selection. In contrast, our method aims to use cardinality as an objective to optimize a set of trajectories in continuous space.
> > (2) The motivation is different. The main motivation behind [1] for using the induced cardinality is that the user engaged set E is different from the recommended set S, causing a train-test mismatch, and MIC aligns better with the user engagement semantics. In our case, we use the cardinality because it is more suitable than NLL loss for continuous optimization with large deep neural networks due to its tolerance of similar samples. As the samples produced by the network can easily become similar during SGD updates, NLL-based approaches often face numerical instability. A concurrent work [2] also noticed this problem with NLL, and has to develop special optimization techniques to address it. We also have shown in our new experiments that using cardinality is able to overcome this problem.
> >
> > [1] Gillenwater, Jennifer A., et al. "Maximizing induced cardinality under a determinantal point process." Advances in Neural Information Processing Systems. 2018.
> > [2] "Deep Learning of Determinantal Point Processes via Proper Spectral Sub-gradient." Submitted to ICLR 2020.
> >
> >
> > ** Distance should be squared in Eq. 8 **
> >
> > That’s indeed a typo and thank you for noticing it. We will fix it in a revised version.

---

### Official Review · AnonReviewer2 · 2019-10-22
**Official Blind Review #2**

**Rating:** 6

**Review:**

This paper presents a novel approach for forecasting object trajectories (e.g., predicted paths of vehicles) forcing diversity of outputs. The authors adopt determinantal point processes (DPPs) to capture the diversity and propose a diversity sampling function (DSF) which consists of a neural network. Its trainable parameter (i.e., \gamma) maps the past trajectory into a set of latent codes and they are decoded to feasible trajectories by a pre-trained conditional variational autoencoder (cVAE). The DSF is trained by maximizing the diversity of the output trajectory. But, since the standard log-likelihood function can be singular, they additionally present an objective function for diversity by maximizing the expected cardinality, which admits replicated outputs. In experiments, the proposed method finds more diverse paths than other competitors under both for synthetic 2D objects and human motions.

This paper is well-written and easy to understand. The contribution can be important as it performs better than other generative networks that are not forcing diversity. Unlike them, the proposed method can capture the diversity trajectories, which requires for safety-critical applications.

Main concerns:

1. The DPP kernel consists of a similarity (equation (8)) and a quality score (equation (9)). However, it is unnatural that the similarity is defined in the data space (x) and the quality is defined in latent space (z). A more naive approach is to define the DPP kernel as a function of x or z. Is there any specific reason to define those scores are defined in different space?

2. To maximize the expected cardinality, it is enough that the eigenvalues of L(\gamma) become large. Does the network find the trivial solution? E.g., all eigenvalues are the same as a very large value. In addition, the proposed diverse loss is not a new approach. The maximum induced cardinality of a DPP was proposed by [1] and its various properties were also studied therein.

3. In a work of [1], a method to optimize the induced cardinality (a similar to diverse loss in this paper) is proposed. And it would be great to compare the maximum induced cardinality (also can be approximated by the greedy algorithm) to the MAP for inference of diverse trajectory.

4. It is also possible to apply DPP MAP inference to a set of latent codes generated from the encoder of cVAE. Then, the decoder can map the diverse latent variables to feasible data trajectories. Are these outputs also diverse? or does the diversity of latent space reflect the diversity in the data trajectory space?

5. Computing the gradient of the proposed diverse loss is expensive since it is the trace of an inverse of a parameterized matrix. How long does it take to learn the proposed DSF compared to other methods?

6. To avoid that the determinants become zero, a popular choice is to shift all eigenvalues with a small amount (this can be done by adding eps * identity matrix to the kernel matrix). Did the authors investigate other practical diverse losses?

7. It seems to be possible to train the parameters of the kernel matrix, i.e., k in equation (8) and \omega in equation (9). Did the author try to learn those parameters?

Overall, this work proposes an approach combined cVAE with a DPP for forecasting diverse trajectories and the empirical results are promising as it outperforms other methods. But, its novelty is incremental and competitors are not actually the models capturing diversity. I vote for a weak acceptance but depending on clarifications on the above concerns in an author response, I would be willing to increase the score.

Minor comments:

1 . Please specify the network architecture of DSF and details on the parameter \gamma.


[1] Gillenwater, Jennifer A., et al. "Maximizing induced cardinality under a determinantal point process." Advances in Neural Information Processing Systems. 2018.

**Experience Assessment:**

I have published one or two papers in this area.

**Review Assessment: Checking Correctness Of Derivations And Theory:**

I carefully checked the derivations and theory.

**Review Assessment: Checking Correctness Of Experiments:**

I carefully checked the experiments.

**Review Assessment: Thoroughness In Paper Reading:**

I read the paper thoroughly.

---

> ### Author Response · Authors · 2019-11-08
> **Response to Reviewer #2 (Part 1)**
>
> Thank you for your review and feedback. In the following, we will address your comments and questions.
>
>
> ** Why similarity and quality defined in different space **
>
> The similarity is defined in the data space because such a similarity metric can reflect the true diversity of the data samples. And since the mapping from the latent space to the data space is represented by a deep neural network, which is highly non-linear, the data diversity cannot be well modeled in the latent space.
> The quality is defined in the latent space because it is harder to compute the analytic likelihood in the data space (this is a common problem for both VAE and GAN). For cVAE, we can only evaluate the variational lower bound approximately with Monte-Carlo sampling. For likelihood-free models like GAN, it becomes even harder. In contrast, it is easy to compute the likelihood of the latent code since it is sampled from a multivariate Gaussian, and given the fact that we treat most major and minor modes equally, we think it is both reasonable and practical to use the latent code likelihood as a quality metric.
>
>
> ** Will the network find a trivial solution **
>
> The diagonal elements of $\mathbf{L}(\gamma)$ are bounded by the base quality $\omega$ so the trace (sum of eigenvalues) of $\mathbf{L}(\gamma)$ is also bounded. We believe the trivial solution you are referring to is when the cardinality reaches maximum and $\mathbf{L}(\gamma)$ becomes a diagonal matrix with elements $\omega$. This means all samples are inside the probable region (the R sphere) since their quality is $\omega$, and the samples are far away from each other because off-diagonal values $S_{ij}$ are zero. This is a desired behavior because it makes the samples both diverse and probable. Therefore, the trivial solution you referred to is actually a desired solution.
>
>
> ** Comparison with MIC **
>
> We will definitely cite [1] in a revised version as it is a great paper and is closely related. We agree that [1] and we both use the cardinality of a DPP as a metric, but we want to point out two important differences:
> (1) The context is different. [1] is focused on the subset selection problem, so it is called maximum *induced* cardinality (MIC). If we understand correctly, the theories and approximation methods developed in [1] are geared toward subset selection. In contrast, our method aims to use cardinality as an objective to optimize a set of trajectories in continuous space.
> (2) The motivation is different. The main motivation behind [1] for using the induced cardinality is that the user engaged set E is different from the recommended set S, causing a train-test mismatch, and MIC aligns better with the user engagement semantics. In our case, we use the cardinality because it is more suitable than NLL loss for continuous optimization with large deep neural networks due to its tolerance of similar samples. As the samples produced by the network can easily become similar during SGD updates, NLL-based approaches often face numerical instability. A concurrent work [2] also noticed this problem with NLL, and has to develop special optimization techniques to address it. We will also show in our new experiments that using cardinality is able to overcome this problem.
>
> [1] Gillenwater, Jennifer A., et al. "Maximizing induced cardinality under a determinantal point process." Advances in Neural Information Processing Systems. 2018.
> [2] "Deep Learning of Determinantal Point Processes via Proper Spectral Sub-gradient." Submitted to ICLR 2020.
>
>
> ** Using MIC instead of MAP inference **
>
> MIC may not be suitable for our application as it requires the set size k to be pre-determined (if any size is ok, the ground set will have the maximum cardinality), which is the case for the recommendation system setting but not for our trajectory forecasting setting, because we do not know in advance how many trajectories are diverse inside the ground set, e.g., sometimes all trajectories are the same but sometimes all trajectories are very different.
>
>
> ** MAP inference on the latent codes **
>
> This is an interesting baseline, and we will show results for it in our new experiments. As we mentioned before, due to the non-linear mapping from latent space to data space, data diversity may not be well reflected in the latent space.
>
>
> ** Speed of DSF **
>
> The speed is similar to NLL, as NLL also needs to do matrix inverse when computing gradients, and we didn’t observe a noticeable difference between speed of these two methods. Additionally, DSF converges quickly and all our experiments take less than one hour to train the DSF, even for the large dataset experiments below.

---

> > ### Author Response · Authors · 2019-11-08
> > **Response to Reviewer #2 (Part 2)**
> >
> >
> > ** Add eps * identity to the kernel matrix for NLL **
> >
> > Thanks for your suggestion. We have used it in our baseline and will show the results in the new experiments below. In short, it is able to stabilize for a small number of samples (10) but still leads to instability for a larger number of samples (50).
> >
> >
> > ** Make k and $\omega$ learnable **
> >
> > It is possible to make k and $\omega$ learnable given that we impose some constraints or regularization on them. Because if k and $\omega$ are unconstrained, the network can make k arbitrarily small and $\omega$ arbitrarily large to achieve high cardinality.
> > In our opinion, it is better to let users control k and $\omega$ as a way to trade-off between diversity and quality. It is also possible to condition the network on k and $\omega$ so users can achieve such trade-off dynamically without retraining the network.
> >
> >
> > ** Large dataset experiments and more baselines (Requested by R2 & R3) **
> >
> > We have further performed more extensive experiments to test the generalization of our method.
> >
> > Dataset: We use a large public human motion dataset *Human3.6M* to perform a new experiment on human motion forecasting. We use past 0.5s to forecast future motions in 2s.
> >
> > Baselines:
> > (1) NLL (requested by R2 & R3):  using the negative log-likelihood of the ground set under DPP as the loss. We also add eps * I to improve its stability.
> > (2) DSF-COS (different kernels as requested by R3):  using cosine similarity as the similarity measure for the DPP kernel.
> > (3) cVAE: random sampling from the Gaussian prior.
> > (4) cVAE-LDPP (requested by R2):  We sample 100 latent codes and use DPP MAP inference on the latent codes to obtain N diverse latent codes, which are decoded into a set of trajectories.
> >
> > Results:
> > For the metrics, lower (ADE, FDE) and larger (ASD, FSD) are better.
> >
> >  Number of samples N = 10
> >  ════════════════════════════
> >                                 ADE     FDE     ASD     FSD
> >  ────────────────────
> >   DSF (Ours)        0.340   0.521   0.381   0.621
> >   NLL                     0.335   0.514   0.343   0.496
> >   DSF-COS            2.588   1.584   5.093   5.718
> >   cVAE                   0.363   0.549   0.235   0.360
> >   cVAE-LDPP        0.373   0.554   0.280   0.426
> >  ════════════════════════════
> >
> >  Number of samples N = 50
> >  ════════════════════════════
> >                               ADE     FDE     ASD     FSD
> >  ────────────────────
> >   DSF (Ours)       0.236   0.306   0.313   0.415
> >   NLL                          X          X          X           X
> >   DSF-COS           0.978   0.891   2.007   1.968
> >   cVAE                  0.276   0.369   0.160   0.220
> >   cVAE-LDPP       0.277   0.365   0.176   0.240
> >  ════════════════════════════
> > *X means that NLL is unable to learn a model due to numerical instability
> >
> > (1) Comparison with a different kernel:
> > The baseline DSF-COS uses cosine similarity for the DPP kernel, we can see that it is able to achieve high diversity but its samples are overly diverse and have poor quality which is indicated by large ADE and FDE.
> >
> > (2) DSF vs NLL:
> > We can see that for a small number of samples (N=10), our method achieves better diversity (larger ASD, FSD) and similar ADE, FSD to NLL. For a larger number of samples (N=50), NLL becomes unstable even with a large eps=1e-3 added to the diagonal. This behavior of NLL, i.e., stable for small N but unstable for large N, matches our intuition that NLL becomes unstable when samples become similar because when there are more samples, it is easier to have similar trajectories. And the reason why NLL cannot handle similar trajectories is that the DPP kernel L becomes singular as discussed in the paper (Section 4.2).
> >
> > (3) DPP MAP inference in the latent space:
> > For the cVAE-LDPP, it is indeed able to generate more diverse samples than cVAE but the diversity is worse than DSF. We think the main reason is that the mapping from the latent space to the data space is highly non-linear, and a diverse set of latent codes won’t necessarily result in a diverse set of samples.
> >
> > ** Network architecture of DSF **
> >
> > We show the architecture of the DSF in Fig. 6 (Appendix B).

---

### Official Review · AnonReviewer1 · 2019-10-23
**Official Blind Review #1**

**Rating:** 8

**Review:**

The authors propose a method to diversify samples generated from a VAE. The method is based on determinantal point processes in the latent space and relies on specifying a kernel in the sample space and a quality metric in the latent space.

Increasing diversity of samples from VAE-like models is an important and common problem and the authors present a reasonably generic method for solving it. The experiments are sensible and show clear improvement over a reasonable selection of baselines. One improvement I would suggest is adding a more explicit discussion of how the proposed method avoids generating overly diverse, very improbable trajectories. It is also somewhat suspicious that the trajectories shown in Figure 3 are terminated so early.

**Experience Assessment:**

I do not know much about this area.

**Review Assessment: Checking Correctness Of Derivations And Theory:**

I assessed the sensibility of the derivations and theory.

**Review Assessment: Checking Correctness Of Experiments:**

I assessed the sensibility of the experiments.

**Review Assessment: Thoroughness In Paper Reading:**

I read the paper at least twice and used my best judgement in assessing the paper.

---

> ### Author Response · Authors · 2019-11-08
> **Response to Reviewer #1**
>
> Thank you for your review and suggestions. In the following, we will address your comments.
>
> ** How to avoid overly diverse and very improbable trajectories **
>
> Our method provides two ways to trade-off between diversity and quality (trajectories that are more probable). (1) We can reduce the $\rho$ or equivalently $R$ in the quality term to force samples to move closer to the origin of the latent space, which results in more probable trajectories. (2) We can reduce the $k$  in the similarity metric to change the scale of the RBF kernel so that the samples have less incentive to move away from each other because the gain from it will be exponentially smaller. To optimize the cardinality, the DSF network will instead focus on moving samples toward more probable regions.

---

### Decision · Program_Chairs · 2019-12-19

**Decision:**

Accept (Poster)

**Comment:**

The paper proposes an approach for forecasting diverse object trajectories using determinantal point processes (DPP). Past trajectory is mapped to a latent code and a conditional VAE is used to generate the future trajectories. Instead of using log-likelihood of DPP, the propose method optimizes expected cardinality as a measure for diversity. While there are some concerns about the core method being incremental in novelty over some existing DPP based methods, the context of the paper is different from these papers (ie, diverse trajectories in continuous space) and reviewers have appreciated the empirical improvements over the baselines, in particular over DPP-NLL and DPP-MAP in latent space.